# Dynamic simulations of feeding and respiration of the early Cambrian periderm-bearing cnidarian polyps

Yiheng Zhang[1,2†], Xing Wang[3*†], Jian Han[2*], Juyue Xiao[1], Yuanyuan Yong[2], Chiyang Yu[2], Ning Yue[2], Jie Sun[2], Kaiyue He[2], Wenjing Hao[2], Tao Zhang[1,2*], Bin Wang[1], Deng Wang[2], Xiaoguang Yang[2]

[1]School of Information Science & Technology, Northwest University, Xi'an, China; [2]State Key Laboratory of Continental Dynamics, Shaanxi Key Laboratory of Early Life and Environments, Department of Geology, Northwest University, Xi'an, China; [3]College of Life Science, Linyi University, Linyi, China

*For correspondence:
wx5432813@126.com (XW);
elihanj@nwu.edu.cn (JH);
zhangtao129@nwu.edu.cn (TZ)

†These authors contributed
equally to this work

Competing interest: The authors
declare that no competing
interests exist.

Reviewing Editor: David
A Paz-Garcia, Centro de
Investigaciones Biológicas del
Noroeste (CIBNOR), Mexico

## eLife assessment

This **important** study advances our understanding of early Cambrian cnidarian paleoecology and suggests that the reconstructed ancestral feeding and respiration mechanisms predate jet-propelled swimming utilized by modern jellyfish. The work combines **solid** evidence of fluid and structural mechanics modeling, simulating for the first time the feeding and respiratory capacities in a micro-fossil (Quadrapyrgites), which in turn opens new possibilities using this approach for paleontological research. Assuming that the prior interpretations and assumptions concerning the modeled organism's soft part and skeletal anatomy are correct, the hypotheses that (1) the organism could alternately contract and expand the oral region and (2) such movement increased feeding efficiency seem plausible.

**Abstract** Although fossil evidence suggests the existence of an early muscular system in the ancient cnidarian jellyfish from the early Cambrian Kuanchuanpu biota (ca. 535 Ma), south China, the mechanisms underlying the feeding and respiration of the early jellyfish are conjectural. Recently, the polyp inside the periderm of olivooids was demonstrated to be a calyx-like structure, most likely bearing short tentacles and bundles of coronal muscles at the edge of the calyx, thus presumably contributing to feeding and respiration. Here, we simulate the contraction and expansion of the microscopic periderm-bearing olivooid *Quadrapyrgites* via the fluid-structure interaction computational fluid dynamics (CFD) method to investigate their feeding and respiratory activities. The simulations show that the rate of water inhalation by the polyp subumbrella is positively correlated with the rate of contraction and expansion of the coronal muscles, consistent with the previous feeding and respiration hypothesis. The dynamic simulations also show that the frequent inhalation/exhalation of water through the periderm polyp expansion/contraction conducted by the muscular system of *Quadrapyrgites* most likely represents the ancestral feeding and respiration patterns of Cambrian sedentary medusozoans that predated the rhythmic jet-propelled swimming of the modern jellyfish. Most importantly for these Cambrian microscopic sedentary medusozoans, the increase of body size and stronger capacity of muscle contraction may have been indispensable in the stepwise evolution of active feeding and subsequent swimming in a higher flow (or higher Reynolds number) environment.

## Introduction

Cnidarians, such as medusozoans, corals, sea fans, and hydromedusae, are generally considered to be a sister group of bilateral animals that live predominantly in the ocean. In general, medusozoans have a two-stage life cycle, consisting of a swimming medusoid stage and a sedentary polypoid stage. Swimming jellyfish rely on rhythmic contraction of the coronal muscles and expansion of the mesoglea at the umbrella rim to swim in a 'jet-like' manner in the water column (*Arai, 1997*; *Brusca et al., 2016*; *Leclère and Röttinger, 2016*; *Zapata et al., 2015*). Because both the ectoderm and endoderm of jellyfish are in direct contact with seawater, no specialised respiratory organs are required to meet aerobic metabolic needs. Sedentary polyps rely on free, extensible tentacles and can feed actively or passively (*Brusca et al., 2016*).

Although the origin of the common ancestor of medusozoans was dated using the molecular clock technique to the Cryogenian (*Erwin et al., 2011*), the earliest known medusozoan fossil records were found in the Ediacaran. *Haootia quadriformis* from the Ediacaran Fermeuse Formation (ca. 560 Ma) was suggested to be a stalked jellyfish based on external morphological evidence and possible coronal muscles on its surface (*Liu et al., 2014*). By comparing periderm morphology, several taxa of tetradial conulariids in the Ediacaran were also proposed to be more closely related to modern scyphozoan polyps (*Van Iten et al., 2006*; *Leme et al., 2022*). All these fossil types were suggested to be sedentary forms, with no definitive evidence of a free-swimming lifestyle. The earliest known swimming jellyfish, *Yunnanoascus haikouensis* (*Hu et al., 2007*), from the early Cambrian Chengjiang biota (Stage 3, ca. 519 Ma) exhibits a typical tetraradial symmetry; eight sensory rods distributed around the umbrella rim, 16 pairs of elongated retractable tentacles evenly spaced with rhopalias, and a less pronounced manubrium. This configuration allows direct comparison with modern scyphozoans (*Li et al., 2007*; *Han et al., 2016a*) and suggests that the origin of swimming jellyfish may have occurred much earlier.

Phosphatised microfossil medusozoans from the early Cambrian Kuanchuanpu biota (ca. 535 Ma) provide critical clues for investigating the origin and evolution of cnidarians and swimming medusae. At least four families have been identified in the Kuanchuanpu biota, involving Hexangulaconulariidae,

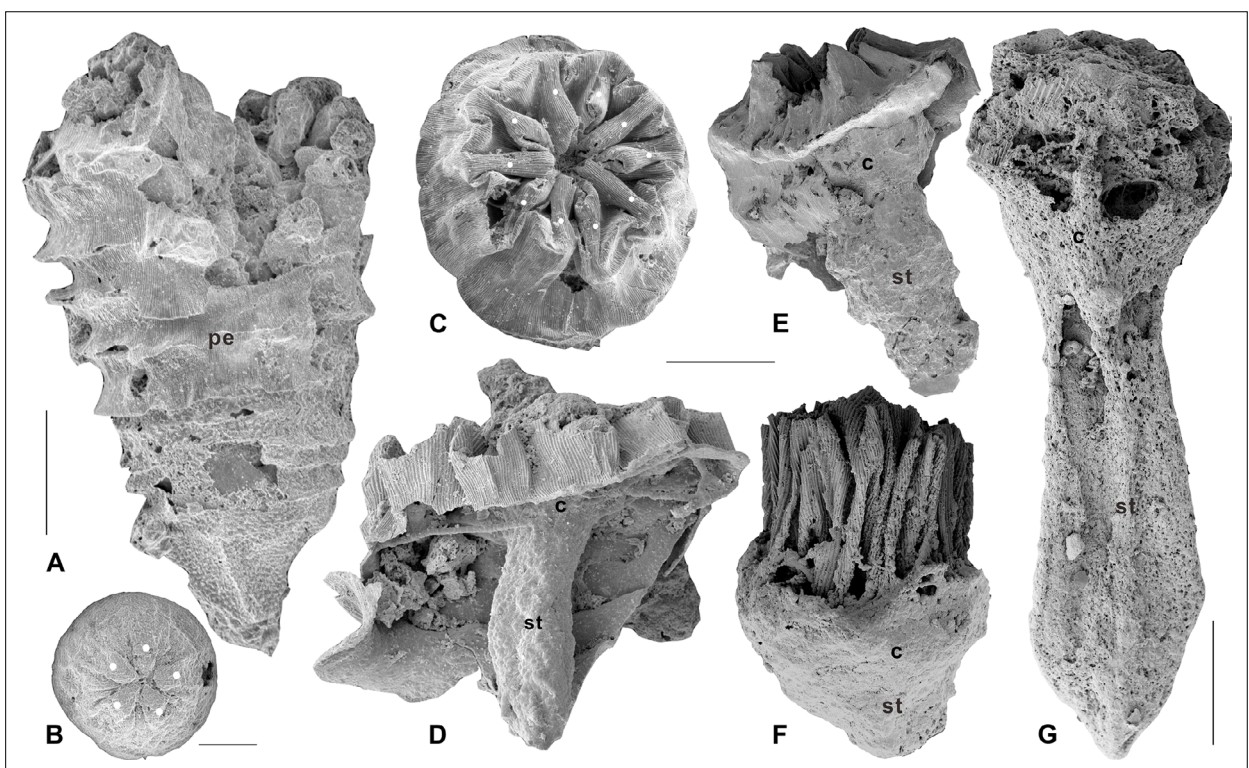

**Figure 1.** Olivooides mirabilis from the Cambrian Fortunian Stage Kuanchuanpu Formation, in Shizhonggou section, Ningqiang County, Shaanxi Province, China. (**A**) Juvenile stage; (**B**) prehatched development stage; (**C**) peridermal apertural view; (**D–G**) possible internal calyx-like polyp. Solid white dots indicate the pentaradial symmetry. Abbreviations: c, calyx; st, stalk; pe, periderm. Scale bars: (**A**) 500 µm; (**B**) 200 µm; (**C–F**) 300 µm; (**G**) 400 µm.

Anabaritidae, Carinachitidae, and Olivooidae (*Han et al., 2020*). Of these, the family Olivooidae includes the tetraradial symmetrical *Quadrapyrgites* as well as multiple pentaradial forms, such as *Olivooides*, *Sinaster*, *Hanagyroia*, and other undetermined taxa (*Li et al., 2007*; *Liu et al., 2014*; *Han et al., 2013*; *Han et al., 2016b*; *Dong et al., 2013*; *Dong et al., 2016*; *Wang et al., 2017*; *Wang et al., 2020*).

As revealed by scanning electron microscopy (SEM), the hemi-globular-shaped embryo of *Olivooides* (*Figure 1*), enclosed by an egg membrane, is equipped with very complex internal structures at the prehatched embryonic stage, such as a manubrium in a relatively deep subumbrella cavity, short tentacles, bundles of coronal muscles, paired gonad-like lamellae at either side of the interradial septa, and many other sheet-like lamellae (*Han et al., 2013*; *Han et al., 2016a*; *Dong et al., 2013*; *Wang et al., 2017*; *Wang et al., 2020*). Remarkably, the regular distribution of ring-like fibrous structures on the surface of the umbrella of their prehatched embryos, which are densely packed in bundles at the edge of the subumbrella and gradually become sparse towards the aboral side, allows for a comparison with the coronal muscles of modern jellyfish (*Han et al., 2020*; *Wang et al., 2022*).

In the hatched stages of development, the soft tissue of millimetre-scale olivooids consists of an upper calyx and basal stalk (*Figure 1G*), in a torch-like shape similar to that of the extant medusozoan polyps (*Wang et al., 2020*; *Steiner et al., 2014*). Unfortunately, the internal structure of polyps remains ambiguous. Considering their cubomedusa-type anatomy in the prehatched stage, the hatched torch-shaped olivooids appear to be a type of periderm-bearing polyp-shaped medusa (*Wang et al., 2020*). As mentioned above, the rhythmic contraction and expansion of the coronal muscles aided by the mesoglea leads to the consequent inhalation and discharge of water, propelling modern jellyfish to swim through the water column, which in turn facilitates more efficient tentacle feeding (*Brusca et al., 2016*). The presence of coronal muscles in early Cambrian embryonic olivooids

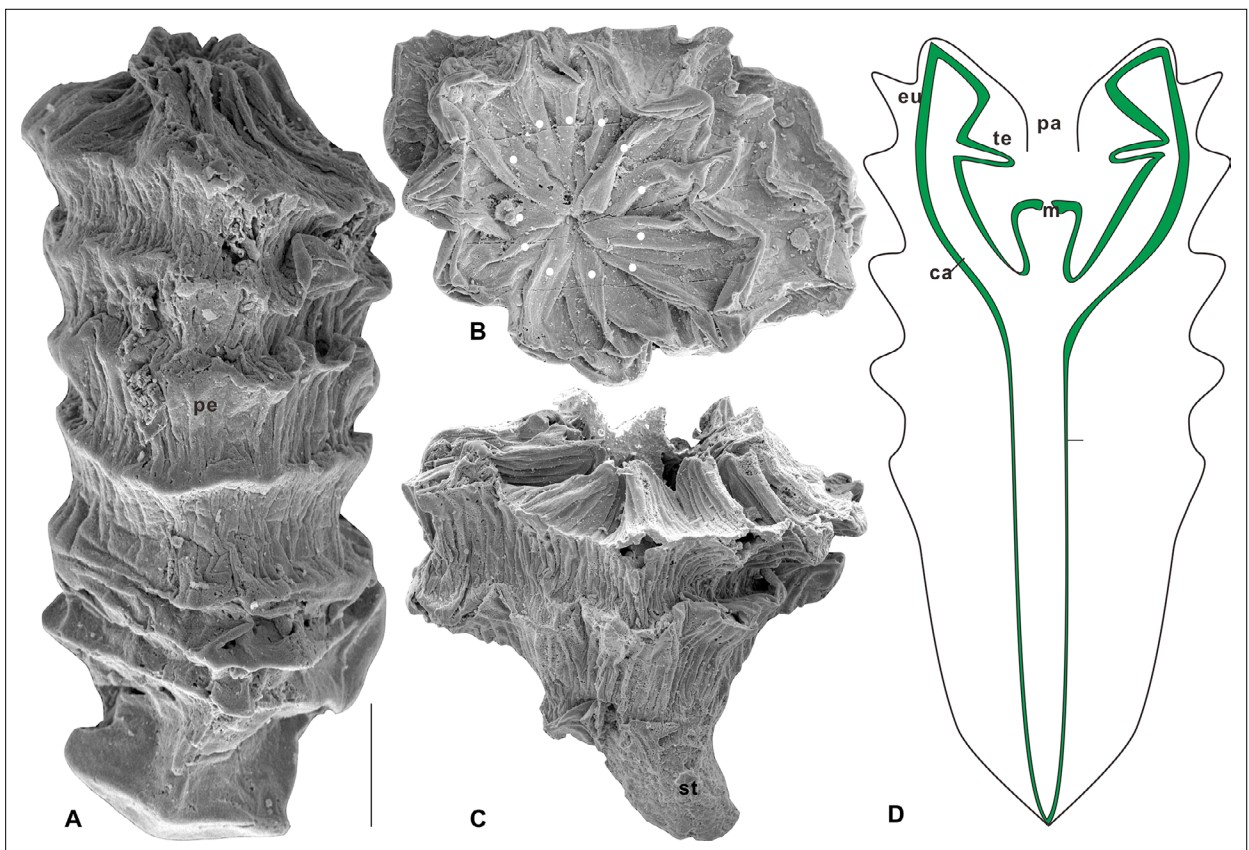

**Figure 2.** Quadrapyrgites quadratacris of the Cambrian Fortunian Stage Kuanchuanpu Formation, in the Zhangjiagou section, Xixiang County. (**A**) Polyp stage with uplifted peridermal aperture; (**B**) apertural view showing concaved peridermal aperture; (**C**) possible calyx-like polyp with a stalk. (**D**) A 2D structure of *Quadrapyrgites* with simplified internal anatomy. Solid white dots indicate the pentaradial symmetry. Abbreviations: eu, exumbrella; su, subumbrella; sc, subumbrellar cavity; te, tentacle; m, mouth; pe, periderm; pa, periderm aperture; st, stalk; ca, calyx. Scale bar: 200 µm.

suggests that early medusozoans in the hatched/juvenile polypoid stage probably used a similar muscular system to control the opening and closing of the subumbrella to drive the water flow in and out, and to assist in feeding (*Wang et al., 2022*) ; however, the early Cambrian polyp-shaped medusa most likely could not swim freely due to the external periderm. The compromise of these two seemingly conflicting conditions led to the hypothesis on the benthic origin of medusa swimming and feeding (*Han et al., 2020*; *Wang et al., 2022*). Specifically, there may be a series of intermediate types (i.e. stalked jellyfish) between sedentary polyps and free-swimming medusae. These transitional types may have evolved divergently from sedentary forms to free-swimming medusae through a series of morphological and structural innovations in evolution, such as rhythmic contraction of the coronal muscles, loss or degradation of the periderm, and increased thickness of the mesogleal layer (*Han et al., 2020*; *Wang et al., 2022*).

*Quadrapyrgites* (*Figure 2A and B*) are one of the most recognisable taxa in the olivooids from the Kuanchuanpu biota (Fortunian Stage, early Cambrian). It has drawn much attention from biologists and palaeontologists, given its tetra-radial symmetry comparable to that of modern jellyfish (*Liu et al., 2014*; *Dzik et al., 2017*). The pagoda-shaped, thin, flexible periderm was divisible into basal and an abapical sections. The abapical section showed an increasing number of annular ridges as it grew. The surface of an annular ridge exhibits many irregular longitudinal folds and fine striations (*Yong et al., 2022*). Similar to the type of pentaradial forms found in olivooids (*Steiner et al., 2014*, Figures 10.3, 11.6, and 11.13), the 12 longitudinal apertural lobes of *Quadrapyrgites* converge towards the central axis of the periderm and then extend downwards, leaving a narrow, star-shaped but contractile channel, which is called the periderm aperture (*Figure 2B*). The upper side of the polyp calyx was bound to the periderm aperture (*Figure 2C*). The manubrium in the subumbrella cavity was conceived with a mouth at the top. A ring of four pairs of short tentacles was possibly located close to the subumbrella margin, as inferred from other contemporaneous tetraradial olivooid embryos (*Figure 2D*, *Han et al., 2016b*). The varying heights and expansions of the peridermal aperture of *Quadrapyrgites* with the 12 centripetal lobes (*Figure 2A–C*) indicate the peridermal aperture could move up and down along the body axis, and expand centrifugally or contract centripetally, a behaviour that was undoubtedly triggered by the interaction of circular and longitudinal muscles and the mesoglea of the polyp inside the periderm. To date, there is no evidence to support that the tentacles of *Quadrapyrgites* could protrude from the periderm to feed in the same way as modern scyphopolyps.

Modelling the living environment of macrofossils to verify their morphological and functional roles is one of the most recent advances in paleobiology (*Dynowski et al., 2016*; *Darroch et al., 2017*; *Waters et al., 2017*; *Gibson et al., 2019*; *Rahman et al., 2020*; *Song et al., 2021*). Ediacara fossil assemblages with complex ecosystems consist of exceptionally preserved soft-bodied eukaryotes of enigmatic morphology, which their affinities are mostly unresolved (*Tarhan et al., 2018*; *Evans et al., 2022*). For example, computational fluid dynamics (CFD) methods were used to simulate oral feeding in the Ediacaran *Tribrachidium heraldicum*, demonstrating that its oral morphology was more oriented towards suspension filter feeding, providing evidence for late Ediacaran ecosystem complexity (*Rahman et al., 2015*). Our recent findings suggest that fluid simulation tools can also be used for microfossil morphological and functional studies (*Liu et al., 2022*). Additionally, compared with macrofossil fluid simulations, the boundary layer conditions should be considered (*Zhang et al., 2022*). Although the swimming mechanism of modern jellyfish has long been studied by biological modelling and the use of fluid simulations (*Sahin et al., 2009*; *Gemmell et al., 2013*; *Gemmell et al., 2018*), to the best of our knowledge, such methodologies have hardly been applied to modelling the dynamic pattern of Cambrian sedentary polyps.

In the present study, we attempted to simulate both the contraction of the coronoid muscle of the subumbrella and the expansion of the mesoglea layer of structurally simplified polyps of *Quadrapyrgites* using a fluid-structure interaction CFD (*Figure 2*). Thus, we were able to reconstruct and investigate the active dynamic pattern of *Quadrapyrgites*. It is also possible to further probe into the autecology of more microscopic Cambrian sedentary periderm-bearing polyps (*Figure 3*).

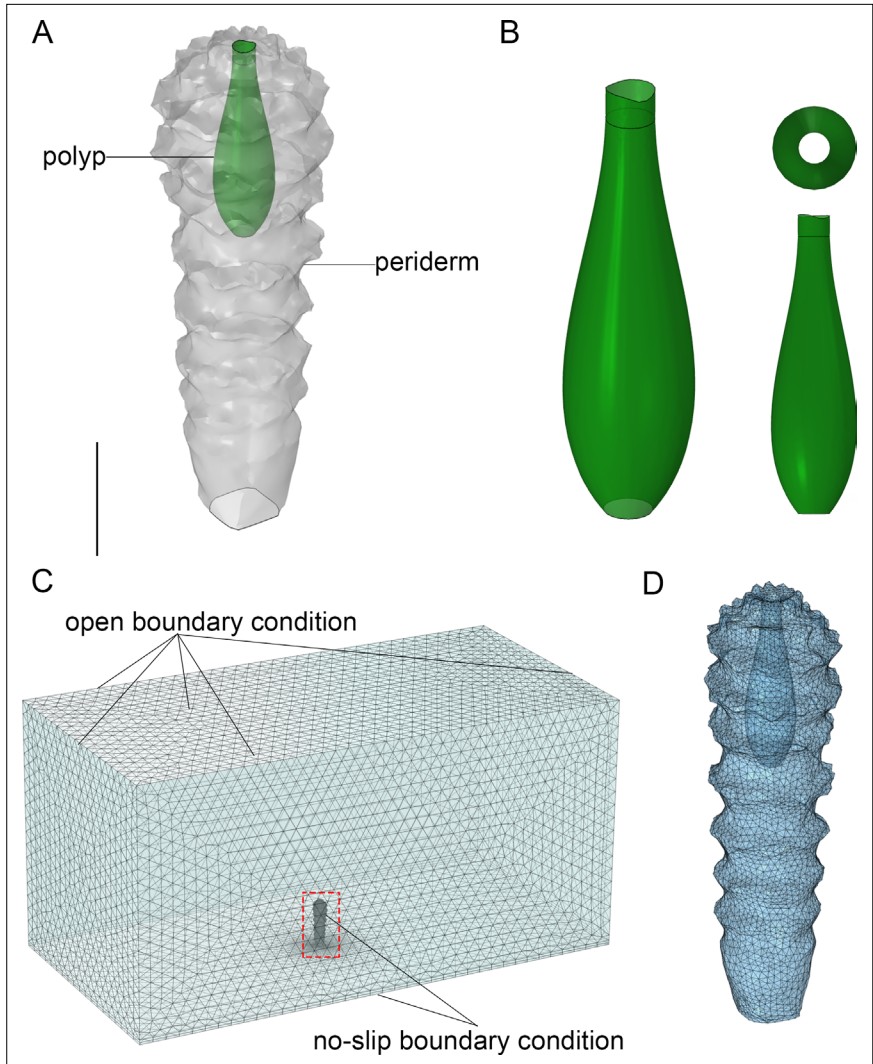

**Figure 3.** Three-dimensional modelling and meshing of *Quadrapyrgites*. (**A**) A 3D model of *Quadrapyrgites;* (**B**) a reduced model of the polyp subumbrella; (**C**) meshed computational domain and boundary conditions (dashed box in **C** marks the position of **D**). Scale bar for (**A**): 200 μm.

The online version of this article includes the following figure supplement(s) for figure 3:

**Figure supplement 1.** Details of model and computational domain settings.

## Results

### Flow velocity

The velocity line profiles for the simulations with different expansion/contraction time ratios show that the velocities in the contraction phase of the four sets of simulations had almost the same trend with time (Table supplement 3 in figshare). During the expansion phase, the maximum values of the mouth flow velocity for all four sets of simulations increased as the expansion velocity increased, with the maximum values occurring to the right of the centre of the time axis (*Figure 4A–D*). By comparing the maximum values of the flow velocities at the sampling cut points in each simulation (*Figure 5*), the accelerated expansion velocity lead to a more remarkable change in flow velocity within the region of z from 2.05 to 2.15 mm than that within the region of z>2.15 mm. Since the flow velocity dropped to below 0.001 m/s, suggesting that the polyp subumbrella had a reduced capacity to take in food from this region.

In all simulations, the general trend of the flow changed with different expansion/contraction time ratios; however, only the maximum values of the flow velocities differed. We considered the results of

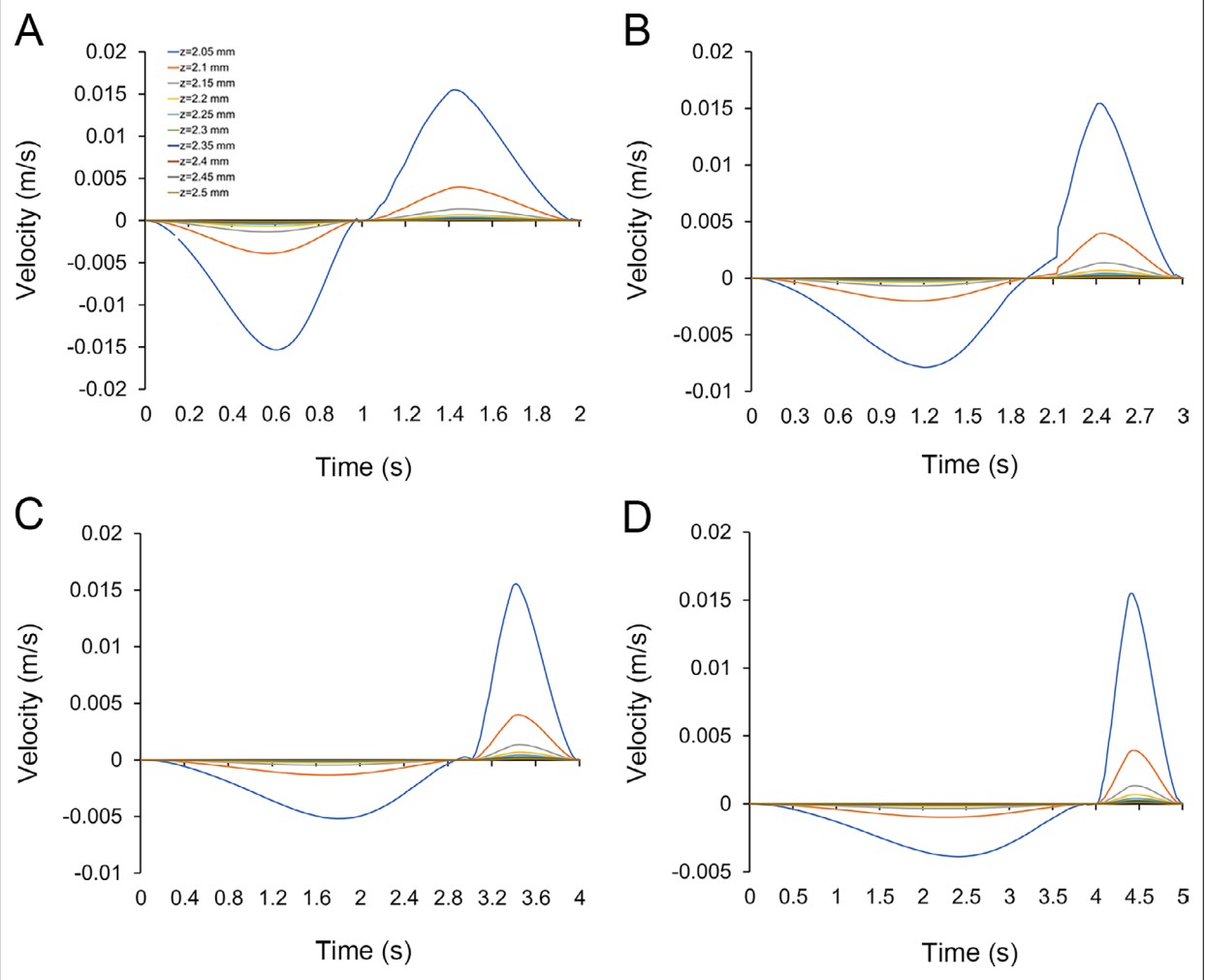

**Figure 4.** Flow velocity profiles of simulations. (**A–D**) are the velocity variations with time collected by sampling cut points in all simulations with expansion/contraction time ratios of 1:1, 2:1, 3:1, and 4:1, respectively.

the simulation with an expansion/contraction time ratio of 3:1 as an example. The subumbrella inside the polyp was in a resting state at 0 s (*Figure 6A and B*), the flow velocity in the flow field was close to 0 m/s, and the opening of the subumbrella was in an expanded state. After 0 s, the subumbrella started to expand, and the external water flow was sucked in. Then, the expansion velocity of the subumbrella gradually decreased with time until around 3 s, when the flow velocity near the opening of the subumbrella became 0 m/s and the mouth shrank to a minimum (*Figure 6E and F*). After 3 s, the subumbrella started to contract, accelerating the contraction, and the opening of the subumbrella began to be restored. At approximately 3.5 s, the contraction velocity of the mesoglea layer reached a maximum, and the flow velocity near the opening of the subumbrella also reached a maximum during the contraction phase (*Figure 6G and H*). At 4 s, the subumbrella stopped contracting, the opening of the subumbrella was restored to its original state, and the flow velocity decreased to a minimum. At this point, the polyp completed the expansion/contraction cycle.

## Vortex visualisation

The visualisation of the intensity of the dimensionless vorticity around *Quadrapyrgites* during its expansion and contraction phases is shown in *Figure 7—animation 1*. The magnitude of vorticity of the colour scale bar was set to [−0.001,0.001] so that the magnitude was appropriate for visualising vortex formation near the periderm. At approximately 0.1 s (*Figure 7A*), the main vortex started to form near the peridermal aperture, and a secondary vortex, which flowed in the opposite direction to the main vortex, also started to form in the middle portion of the periderm. Both the

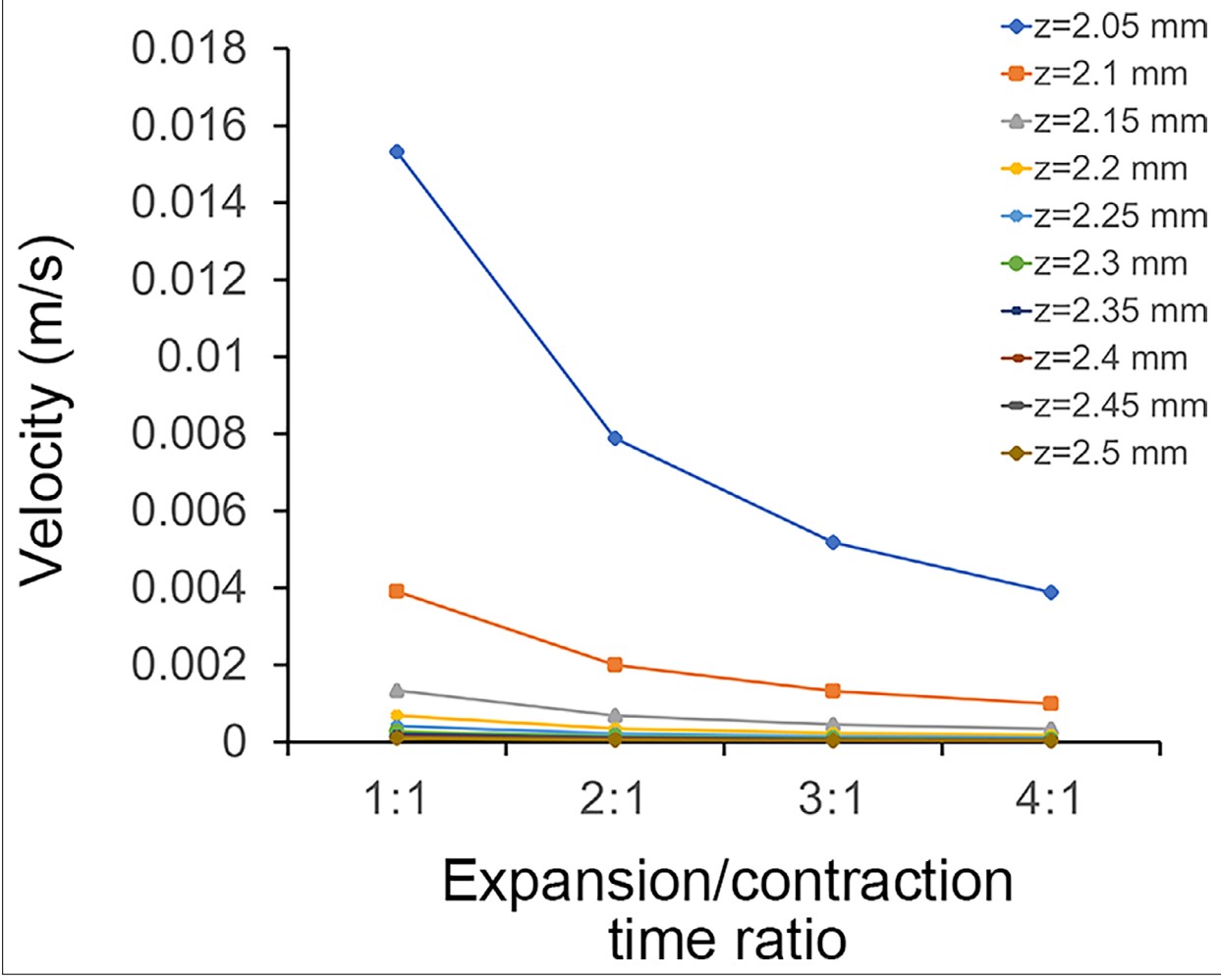

**Figure 5.** Maximum flow velocity data collected by sampling cut points in all simulations with different expansion/contraction time ratios.

main vortex and secondary vortex enlarged gradually over time. At approximately 1 s, the lower secondary vortex was in contact with the bottom surface (*Figure 7B*), and its morphology changed, manifesting itself as a flow from the middle of the periderm towards the bottom surface. At approximately 2 s, the main vortex developed to the maximum visualisation range (*Figure 7C*), at which time, due to an increase in the velocity of the water, a partial microflow in the opposite direction of the vortex was also formed on the surface of the periderm. After 2 s, the secondary vortex began to move up along the surface of the periderm to a position close to the aperture. From 2.87 to 2.88 s (*Figure 7D and E*), the main vortex separated from the periderm, and the secondary vortex moved to the original position of the main vortex at the peridermal aperture, replacing it with the main vortex for the next stage of contraction at the end of the expansion movement of the subumbrella at 3 s (*Figure 7F*).

After the onset of contraction, the newly formed main vortex pushed the expansion phase vortex away from the peridermal aperture. The vortex development process in the contraction phase follows a pattern similar to that in the expansion phase, with the vortex eventually separating from the peridermal aperture at 3.9–4.0 s (*Figure 7G and H*), at which point the expansion/contraction cycle was completed.

Because the contraction process took less time and the vortex enlarged faster than that in the expansion phase, the shapes and maximum sizes of the newly formed main vortex and secondary vortex differed, but the overall trend of alternating main vortex and secondary vortex formation was maintained during the expansion-contraction-expansion movement of the subumbrella.

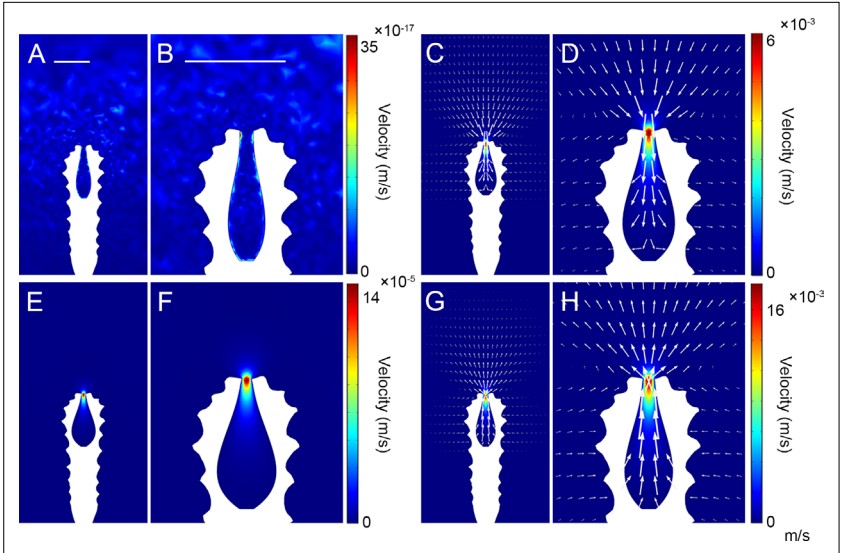

**Figure 6.** 2D velocity visualisations of the simulation with expansion/contraction time ratio of 3:1. (**A, C, E, and G**) show the status of the polyp subumbrella at different moments. (**B, D, F, and H**) show the corresponding enlarged views of the polyp, with the white arrows representing the flow direction and velocity magnitude (the size of the arrows is proportional to the natural logarithm of the flow velocity magnitude with a range quotient of 1000) of water flow. (**A, B**) t=0 s, the polyp is at rest, and the subumbrella opening is in its maximum state. (**C, D**) t=1.8 s, the subumbrella is in the process of expansion, and the flow velocity near the peridermal aperture has reached its maximum. (**E, F**) t=3 s, the subumbrella is in its maximum state, and the subumbrella opening is in its minimum state. (**G, H**) t=3.5 s, the subumbrella is in the process of contraction, and the flow velocity near the peridermal aperture has reached its maximum value. Scale bar: 500 μm.

## Discussion

### Expansion/contraction frequency and feeding and respiration efficiencies

The simulations demonstrated the dynamic pattern of *Quadrapyrgites* during one cycle of expansion and contraction and the visualisation of ambient water flow in its vicinity. A faster expansion rate (i.e. a shorter expansion-contraction cycle) leads to a relatively greater water exchange and flow velocity formed near the subumbrella aperture. Subsequently, the tentacles will have more opportunities to make contact with suspended food particles in fresh water inputted from outside the periderm per unit of time. In this regard, compared with the stagnant condition, the increased velocity of the water flowing into the periderm and then the subumbrella cavity will improve the efficiency of food intake. In the subsequent contraction phase, the polyp expels water from the subumbrella cavity at a high rate of movement.

During the contraction/expansion movement of the polyp, the vortices formed around the periderm could slowly bring food particles close to the periderm aperture, where small food particles were more likely to gather owing to the viscous force of the peridermal surface instead of being transported away by the current. This combination of active and passive feeding could also improve the feeding and gas exchange efficiencies of polyps. Notably, although the deeply concave subumbrellar cavity of olivooids is equipped with short tentacles, it is unlikely that they can protrude their short tentacles out of the periderm. Therefore, except for the random passive flow of microscopic food particles, their main mode of food intake is likely suspension feeding through active contraction by the ring muscles (*Wang et al., 2022*). The relatively high rate of contraction of *Quadrapyrgites*/olivooids, if contracted frequently or rhythmically, helps the four/five pairs of short tentacles capture relatively larger quantities of food particles in a short period of time. In this regard, olivooids can be functionally considered as active suspension feeders rather than conventional predators.

The feeding and excretory activities of marine benthic organisms are closely related to their surrounding water as an appropriate flow velocity can increase feeding efficiency (*Pratt, 2008*). Active suspension feeders with the ability to move can actively avoid areas where currents are unsuitable for

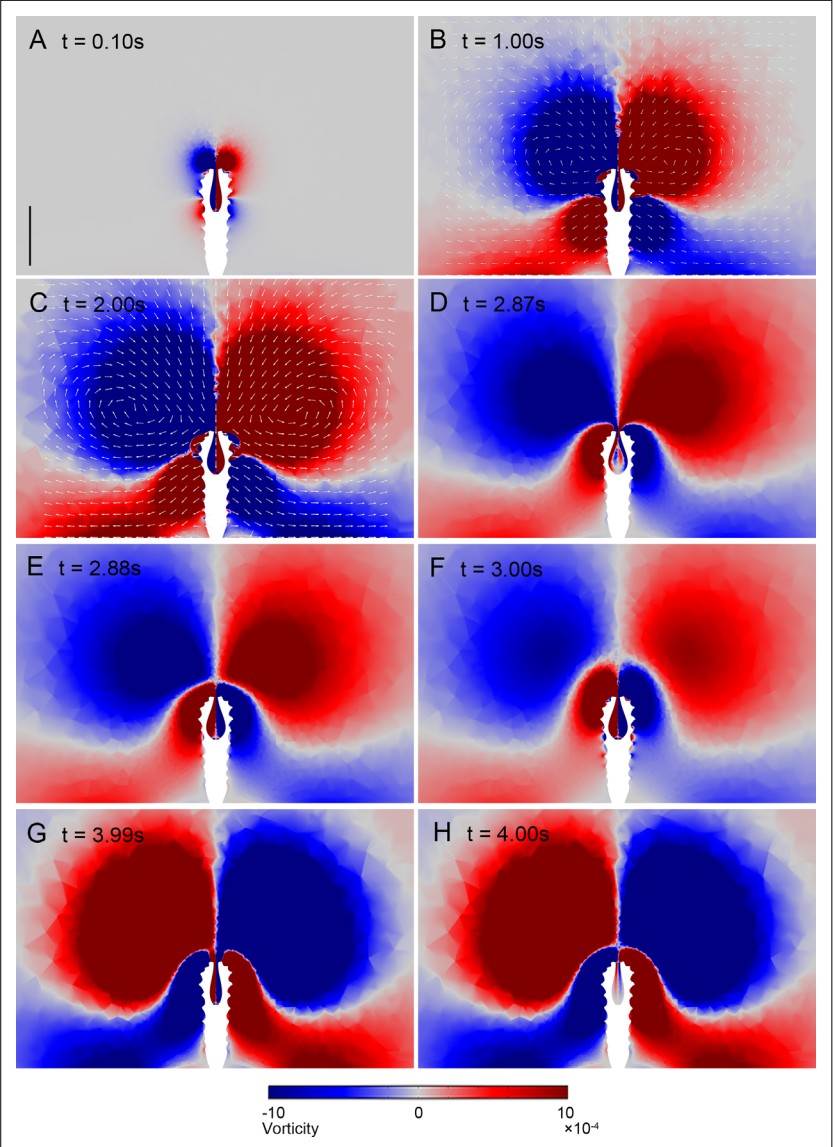

**Figure 7.** Vortex visualisation of the dynamic process of *Quadrapyrgite* (the length of arrows was normalised to represent the orientation of velocity). (**A**) t=0.1 s, the upper main vortex, and the lower secondary vortex begin to form. (**B**) t=1 s, contact between the secondary vortex and the lower boundary. (**C**) t=2 s, the main and secondary vortex is developed to the maximum visualisation range of vorticity. (**D, E**) t=2.87–2.88 s, separation of the main vortex occurs. (**F**) t=3 s, the new main vortex of the contraction process formed, the expansion process ended, and the contraction process began. (**G, H**) t=3.9–4 s, separation of the main vortex occurred. Scale bar: 1000 µm.

The online version of this article includes the following video for figure 7:

**Figure 7—animation 1.** 2D visualisation of vortex (the length of arrows was normalised to represent the orientation of flow velocity).

survival and choose areas where food is abundant (*Labarbera, 1984*). In contrast, in sedentary forms, olivooids were unable to displace directionally and actively; therefore, their feeding was more dependent on the surrounding environments. A suitable current environment is conducive to the formation of eddies around the periderm to inhale food particles efficiently from the space above the subumbrella aperture, whereas excessive current velocities may take away large amounts of food particles.

## Potential influence of current velocity on feeding patterns

It has been suggested that microbenthos mostly inhabited in the low flow region above the seafloor, also called the bottom boundary layer (*Trowbridge and Lentz, 2018*). The flow velocity

above the *Quadrapyrgites* peridermal aperture showed that the maximum velocity ranged from 0.005 to 0.0155 m/s depending on the rate of contraction and expansion (*Figure 5*). Such a flow velocity magnitude indicated that polyps may live in relative low flow environment protected by the viscous boundary layer, which enable them to maintain a relatively stable posture for feeding activities in a lower ambient flow speed environment (*Liu et al., 2022*). Otherwise, inhaling food from the ambient environment with a current velocity much higher than the inhaling velocity is more difficult.

However, the protective ability of the viscous boundary layer is varied because of the fluctuations of flow caused by oscillating current or uneven seafloor (*Zhang et al., 2022*). Simultaneously, the current velocity (at a height of 2–2.5 mm) in the vicinity of the peridermal aperture within the viscous boundary layer might be close to 0.03 m/s (*Caldwell and Chriss, 1979*; *Wengrove and Foster, 2014*), about twice the maximum inhalation velocity of the polyp in the simulations, especially in areas with frequent intertidal current velocity changes (*Wengrove and Foster, 2014*). Under these relatively high current velocities, food particles will be rapidly transported, making it difficult for polyps to inhale them efficiently. Thus, it can be hypothesised that the physiological activity of *Quadrapyrgites* relies on other potential environmental factors in addition to the protectiveness of the viscous boundary layer. For example, the greater viscous effect of rougher sediment surfaces or the enrichment of organisms in the area results in more turbulence among the community, increasing the thickness of the boundary layer (*Grant and Madsen, 1986*), and simultaneously reducing the current velocity. A turbulent environment can enhance the mixing of nutrients, thereby increasing the feeding efficiency (*Denny, 1988*), but the effectiveness requires further investigation. In addition, this type of inhalation feeding is profoundly dependent on the surrounding environment and may be an important constraint in survival.

The pattern of alternating ring-shaped main and secondary vortices formed during the expansion and contraction of polyps (*Figure 7*) shares some similarities with those in modern swimming medusae. Modern medusae swimming patterns can be divided into jet propulsion (jet propulsive force is generated by the contraction of the circular muscle fibres) and jet-paddling propulsion (the edge of the subumbrella can act as 'paddles' to assist jet propulsion) (*Dabiri et al., 2007*). The dye visualisation on *Aurelia aurita* showed a ring-shaped starting vortex generated during the swimming stroke (contraction), followed by a stopping vortex of opposite rotational sense during the recovery stroke (expansion) (*Dabiri et al., 2005*). The vortex formed was suggested to be related to both feeding and propulsion. *Sahin et al., 2009*, recovered the swimming mode of hydromedusae *Sarsia tubulosa* (jet propulsion) and *Aequorea victoria* (jet-paddling propulsion) using a CFD method and visualised the ring-shaped starting and stopping vortices. The flow pattern of a vortex can be analysed to estimate the efficiency at which a jet produces thrust (*Sahin et al., 2009*). *Dabiri et al., 2005*, suggested that the interaction between the starting and stopping vortex functions to reduce the kinetic energy was lost during medusa swimming, while *Sahin et al., 2009*, suggested that the formation of toroidal vortex rings in the wake of medusa swimming does that. Gemmell et al. also suggested that the starting and stopping vortices may be related to the passive energy recapture mechanism, which is supported by the fact that *A. aurita* is one of the most energetically efficient propulsors (*Gemmell et al., 2018*; *Gemmell et al., 2013*). Although the dynamic process of sedentary *Quadrapyrgites* is more or less similar to the jet propulsion swimming of medusae, they may not stroke to move or swim concerning the external periderm. The formed main and secondary vortices are presumed to improve the efficiency of water flow in and out of the polyp cavity, hence influencing the feeding and respiration efficiency. Furthermore, the contraction (or changeable diameter) of the subumbrella opening can bolster the kinetic energy of expelled water and the vortex (*Mohseni and Gharib, 1998*; *Mohseni et al., 2001*), which may also help them gain extra feeding and respiration efficiency. However, jet propulsion is significantly less efficient than jet-paddling propulsion swimming patterns, although the former can produce relatively higher thrust and kinetic energy in most cases. Constrained by peridermal structure, the *Quadrapyrgites* polyps were unable to expand and contract to a large magnitude as modern medusae or hydrae do, thus their dynamic efficiency is considered to be rather limited, although this cannot be calculated by using formulas for modern swimming animals (e.g. net cost of transport analysis proposed in *Gemmell et al., 2013*) due to their sedentary life and the ambiguity of certain truly essential parameters (e.g. net mass of polyps).

## Functions of muscular system in post-embryonic olivooids

Extant anthozoans, and possibly, medusozoan polyps (*Arai, 1997*), are exclusively passive opportunists in feeding (*Shick, 1991*), minimising the energetic cost of obtaining food. In contrast, jellyfish are undoubtedly active feeders (*Arai, 1997*). The earliest Cambrian medusozoan olivooids, although encased in the periderm, may be able to switch between active and passive feeding modes as modern benthic organisms. In an energetic tidal setting with a high concentration of oxygen, despite the low flow due to the bottom boundary layer, olivooids can presumably stay inside the periderm, waiting for the suspension of food particles transported by the bimodal current or absorbing dissolved organic matter. Morpho-anatomical studies suggest that the coronal muscles of the pentaradial symmetrical medusozoan embryos from the early Cambrian Kuanchuanpu biota during the pre-hatching stage are perfectly comparable to those of modern medusae, and it was further hypothesised that the contraction of the coronal muscles of the soft tissue and the expansion of the mesoglea layer guided the opening of the periderm, thereby facilitating animal feeding (*Wang et al., 2022*). Taken together with CFD simulation, it suggested that Cambrian olivooids could also be active feeders and that they may have developed some of the behavioural capabilities of swimming medusae.

## Body size, Reynolds number, and swimming

Previous studies have revealed that some Cambrian echinoderms may adopt a more favourable posture in relation to the current to reduce drag or create a suitable recirculation environment to improve feeding efficiency (*Rahman et al., 2020*). Olivooids were unable to do so because of their sedentary life within a tetraradial symmetrical periderm. However, their periderm aperture will stand upright or tilt at various angles. As polyps and their periderm became larger or taller, due to growth or evolution, the upper portion of the periderm left the bottom boundary layer region (*Zhang et al., 2022*) and encountered higher velocity currents. Therefore, larger polyps have to develop a series of novel strategies to adhere to the seafloor by a holdfast and to access more suspended food particles per unit time transported by higher velocity currents (*Shick, 1991*). One of the strategies for periderm-dwelling medusozoans is to develop stronger coronal muscles to achieve a greater contraction or expansion ability (i.e. higher rate or larger magnitude for contraction and expansion), so as to change the direction of flow to a greater extent. Alternatively, they may have evolved many longer, extensible tentacles protruding from the periderm, which should result in a larger periderm aperture even without a cover, similar to living scyphopolyps (*Jarms et al., 2002*). Third, if the periderm largely degenerated, such as in living sea anemone or cubozoan polyps (*Straehler-Pohl, 2017*), the polyps could change their adhesive position. Otherwise, they could develop a streamlined body to reduce drag and allow for more favourable feeding gestures (*Liu et al., 2022*).

Furthermore, from the aspect of biological fluid mechanics, a low Reynolds number (*Re*) swimming strategy can lead to a lower propulsion efficiency in modern propulsive swimming animals, such as swimming medusae (*Sahin et al., 2009*). In certain cases, animals are unable to move forward if the swimming *Re* is too low; this is called 'the scallop theorem' (*Robertson et al., 2019*). Specifically, this implies the medusae may evolve to much larger sizes and still be able to move through the surrounding fluid (*Sahin et al., 2009*). For modern scyphomedusae, such as *A. aurita*, an increase in body size (or bell diameter) allowed them to swim in a higher *Re* manner (*Feitl et al., 2009*). Simultaneously, for millimetre-scale olivooids inhabiting the viscous boundary layer region, both the dynamic *Re* (approximately 1; calculated from the results in this study, see 'Computational fluid dynamics') and the environmental *Re* could be very low (due to the higher fluid viscosity caused by enhanced mixing of sediments and lower flow), the limitation to the efficiency of food inhalation and respiration through contraction and expansion may also be significant. Thus, it is reasonable that both larger body size and stronger capacity of body contraction of Cambrian polyps may have been indispensable towards the stepwise evolution of active feeding, enabling them to live and subsequently swim in a higher flow (or higher *Re*) environment.

Notably, from the earliest Cambrian to Cambrian Stage 3, the trend of increasing body size was remarkable on a global scale (*Zhuravlev and Wood, 2020*), although with some exceptions, i.e., the large, skeletonised conulariid-like *Paraconularia* found in the terminal Ediacaran Tamengo Formation of Brazil (*Leme et al., 2022*). For example, in South China, millimetre-scale, sedentary medusozoan polyps from the Cambrian Fortunian Stage to Stage 2, except for the tubulous Anabaritidae without tube closure (*Liu et al., 2017*; *Guo et al., 2021*; *Guo et al., 2020b*; *Guo et al., 2020a*), have small

peridermal apertures. For polyps living inside the periderm, access to food is relatively inefficient, and food particles are limited in size by virtue of contraction and expansion of the peridermal aperture and the coronal muscle-mesoglea layer. In contrast, centimetric polyp-type sedentary tube-dwelling cnidarian fossils such as *Sphenothallus* (*Li et al., 2004*), *Cambrorhytium* (*Conway Morris et al., 2015*), and *Byronia* (*Zhu et al., 2000*; *Chang et al., 2018*) from the mid- to late Cambrian allow their tentacles to protrude completely out of the cone-shaped closureless tube. In addition, the tube surface was much smoother, capable of reducing the drag-to-water flow. This body structure, together with protruded tentacles, allowed for larger amounts and sizes of food to be obtained, thus establishing the evolutionary foundation for the rise of polyp strobilation, the emergence of saucer-like planula larvae, and the origin of swimming medusae. Changes in the periderm/exoskeleton of Cambrian benthic medusozoans were also consistent with our hypothesis regarding the interaction between animal body size and current velocity (or *Re* number).

In summary, the simulation results illustrate that the rate of water intake near the periderm aperture is directly related to the expansion rate of the mesoglea layer. Increasing the strength of the expansion-contraction requires a highly concentrated and well-developed coronal muscle and thicker mesoglea layer, which inevitably reduces the density of the polyp body. Hence, olivooid-type feeding was most likely one of the prerequisite transitional forms for the rise of the jet-propelled swimming style; in other words, rhythmic jet-propelled swimming is most likely a by-product of occasional/frequent olivooid-type feeding of periderm-bearing sedentary medusozoans. These inferences fit well with the appearance of centimetre-scale predatory swimming medusae with rhopalias at the beginning of the Cambrian Stage 3 (*Cartwright et al., 2007*; *Han et al., 2016a*).

## Perspectives for future work and improvements

Previous palaeontological CFD simulations applied to extinct fossils have mostly used static models of organisms and have mainly focused on the hydrodynamic efficiency of organisms in water flowing at different velocities (*Gutarra and Rahman, 2022*). This type of simulation has potential for testing hypotheses in terms of an organism's functional and morphological performance (*Rahman, 2017*), providing assessment of and deep insights into the adaptability of organisms to their environment from hydrodynamic perspectives. However, they did not involve the dynamic effects of organisms on the ambient environment. Most previously simulated palaeontological organisms were on the scale of millimetres to centimetres (*Waters et al., 2017*; *Gibson et al., 2019*; *Rahman et al., 2020*; *Song et al., 2021*). The smaller size of *Quadrapyrgites* indicated that they lived in a different current environment compared with larger or taller organisms. For instance, the velocity gradient in the bottom boundary layer flow regime can lead to higher ambient flow speeds with the increasing height of benthic organisms (*Gibson et al., 2021*). Therefore, lack of an ambient current environment may have an impact on the results. However, adding an ambient current to the simulations can introduce more technical issues that are too problematic to be addressed at this stage, such as poorer simulation convergency. In addition, the dynamic effect of *Quadrapyrgites* on ambient water, which was the main focus of the present study, needs to be investigated and visualised without the interference of current (as many CFD simulations for modern jellyfish, e.g. *McHenry and Jed, 2003*; *Gemmell et al., 2013*; *Sahin et al., 2009*, were conducted under hydrostatic conditions). Considering the two important points stated above, the simulations here were conducted under a hydrostatic environment. We emphasised that members of *Quadrapyrgites* were not configured to live in stagnant water naturally, as discussed in a study by *Liu et al., 2022*. However, further investigations of the impact of ambient currents on the feeding abilities of organisms may need to be performed by designing a set of new insightful simulations.

Furthermore, it remains obscure whether *Quadrapyrgites* lived primarily in solitary or gregarious modes. A benthic community with variable organism density can affect ambient water conditions or the feeding capability of a single organism among it (*Gibson et al., 2019*; *Liu et al., 2021*). Remarkably, simulations of large gregarious communities are restricted here by the computational resources and complexity of the model. Although simulating the gregarious benthic communities generally involves modelling a multiple of organisms, which is beyond the present computational capability, the results and data collected here could be used for further simulations to achieve a better understanding of gregarious active feeding and respiration behaviours of the co-existing small shelly fossils. One promising approach is building a simplified active feeding model by reconstructing the flow

velocity profile collected accordingly. Thus, it is possible to further investigate the ecological characteristics of gregarious organisms and the effect of community size on the adaptability of organisms living in the ambient environments.

It is also noteworthy that we omitted the possible effects of the polyp stalk of *Quadrapyrgites* and other internal structures of the calyx on the water flowing in and out of the peridermal aperture. The interaction between these internal structures and subumbrella contraction remains unclear. To this end, we did not model the internal structure of the polyp. In addition, the manubrium in the subumbrella cavity as well as the tentacles could act as a barrier to water flow in and out. The true thickness of the mesoglea is also unknown because of diagenesis, which may influence the exact magnitude of contraction and expansion. However, based on the fossil record, the mesoglea would have been much thinner than that in modern medusae (*Han et al., 2016a*). Although the primary dynamic pattern of *Quadrapyrgites* could be much more subtle, the simplified model required less computational resources and adequate restoration of the polyp body plan. Our study will shed new light on the autecology of Cambrian microbenthos using numerical computational methods.

## Conclusions

Our simulations of *Quadrapyrgites* show that the accelerated expansion of the polyp body can improve active feeding efficiency and increase the range in the upper flow above the peridermal aperture height. The contraction/expansion pattern of the polyp body and rough peridermal surface helps the polyp to access food particles in the ambient environment of the periderm, thereby enhancing the polyp's feeding and gas exchange efficiencies under relatively low flow velocity conditions. Eventually, as body size and height of the Cambrian benthic medusozoans increase, this mode of feeding will be replaced by more efficient feeding methods (e.g. relying on free tentacles). Our study has implications for understanding the feeding and respiration of olivooids and other sedentary medusozoans completely dwelling in their periderm. Furthermore, our findings provide valuable insights into the interactions between the evolution of animal body size, an increased incidence of swimming behaviour in medusa, and the ambient environment during the Cambrian explosion. This is also the first time a dynamic numerical simulation method has been applied to a microfossil, demonstrating the further possibilities for utilising this approach in palaeontological research.

## Materials and methods

### Geological setting and fossil pre-treatment

Rock samples of olivooids were collected from the Shizhongguo and Zhangjiagou sections of the Kuanchuanpu Formation, Shaanxi Province, China. The Shizhongguo section of the Kuanchuanpu Formation is an interbedded set of cherts, flint, and phosphatic tuffs and is approximately 60 m thick. Overlying the Guojiaba Formation is a black carbonaceous shale and siltstone, approximately 8 m thick, dominated by detrital dolomite. The Zhangjiagou in the Kuanchuanpu Formation in section is a thickly bedded set of phosphorus limestone, approximately 22 m thick, whereas the underlying Dengying Formation is dominated by massive, thickly bedded, black dolomite. Small shell fossil specimens were obtained from the *Anabarites-Protohertzina-Arthrochitie*s biozone, which corresponds to the Cambrian Fortunian Stage (*Qian, 1977*; *Qian, 1999*).

The rocks were smashed to a width of 2–3 cm, immersed in a 7–10% acetic acid solution to decompose, and the residue was air-dried before the fossil samples were manually picked out under a binocular microscope (Leica M20 stereoscopic microscope). SEM (FEI Quanta 400 FEG SEM) was used for the scanning photography.

The collected fossil data were modelled using Dragonfly 4.0 and high-resolution 3D models were generated using Autodesk Maya 2018 and saved in the '.stl' format. All fossil specimens and model files were stored at the Shaanxi Key Laboratory of Early Life and Environments and Department of Geology, Northwest University.

### Three-dimensional modelling

Simulations were carried out for *Quadrapyrgites* (*Figure 3—figure supplement 1A*). The 3D model of *Quadrapyrgites* (height: 2.0 mm, length: 0.6 mm, width: 0.6 mm) consists of two parts: an outer pagoda-shaped periderm and a hollow bowl-shaped polyp (*Figure 3A*). The latter is functionally a

proxy of the polyp subumbrella without the manubrium, internal tissue of polyps, and basal stalk (*Figure 3B*). The dynamic process of the simulated polyp was modelled by rhythmic contraction and expansion of the subumbrella, and it was divided into contraction, expansion, and normal resting phases. In the initial state, the subumbrella was about 0.7 mm in height and 0.1 mm in diameter at its widest point. At this point, the umbrella muscle and mesoglea layer contracted, and the size of the subumbrella opening was minimal. Subsequently, the umbrella expanded and increased in size. Finally, the polyp subumbrella shrank and returned to its original shape.

## Computational fluid dynamics

We used COMSOL Multiphysics v. 5.6 (https://cn.comsol.com) to carry out 3D simulations of *Quadrapyrgites*. The computational domain consisted of a cuboid with a length, width, and height of 20, 10, and 10 mm, respectively. The *Quadrapyrgites* model was placed at the centre of the lower boundary of the cuboid domain in the direction of the peridermal aperture-apex from top to bottom and was inserted vertically into the lower boundary approximately 0.18 mm (*Figure 3—figure supplement 1B*). The part below the lower boundary was then removed by Boolean operations (*Figure 3C*). The mesh mainly consisted of the computational domain mesh and boundary layer mesh applied to water-solid interacted boundaries (with a layer number of 2 and stretching factor of 1.2) (*Figure 3A*). A free tetrahedral mesh was used to cover the entire simulation domain to capture as much detail of the model as possible (*Figure 3C and D*). Each subdomain of the whole simulation domain was meshed with specific levels of refinement (i.e. varied maximum and minimum element sizes were applied according to sizes of subdomains) to ensure a balance between the accuracy and computational cost. Sensitivity tests were performed to determine the optimal settings for the subsequent simulations (see 'Mesh sensitivity analysis').

The simulations used the hyper-elastic material model in the membrane node of solid mechanics to define the structure of the umbrella surface of the *Quadrapyrgites* polyp inside the periderm. The stalk and other internal structures of the calyx, such as the manubrium and tentacles in the subumbrella cavity, were ignored in the simplified model of the *Quadrapyrgites* polyp in this study. The subumbrella, with a circular muscle bundle and mesoglea, was replaced with an elastic membrane. Considering that the physical parameters of the polyp umbrella were difficult to obtain directly from preserved fossil material, we used the physical parameters of elastic rubber instead (Odgen material model [*Holzapfel, 2002*], specific parameters are listed in Table supplement 1 in figshare). As the dynamics of the polyp subumbrella were determined by a displacement function and not by the material elasticity of the subumbrella itself, this alternative setting would not have a significant effect on the locomotion of the polyp subumbrella. The boundary of the simulated polyp apex was set as a fixed constraint boundary that supported the simulated polyp subumbrella.

To define the contraction and expansion motion of the umbrella of the polyp using the prescribed displacement method, a columnar coordinate system was established in advance with parameters a, φ, and r, where a is the axis of the centre of rotation of the polyp, φ is the angle of rotation, and r is the distance between the polyp surface and origin. To simplify the physics setup and mathematical model, our simulation reduced the motion of the umbrella inside the polyp to motion in the r-direction only, without considering its motion in the a-direction. For this reason, the displacement of the contraction and expansion motion of the umbrella inside the simulated polyp in the r-direction was defined by an interpolation function with height z as a variable (Table supplement 1 in figshare), and the function was fitted using the cubic spline method to ensure the smoothness of the displacement process inside the simulated polyp.

The ratio of contraction to expansion times varies between different species and even within the same individual in modern cnidarians, with generally short contraction times and relatively long expansion times, as demonstrated by *A. aurita* (*McHenry and Jed, 2003*). Considering that the muscle contraction capacity of the polyp subumbrella may differ from that of modern jellyfish, the contraction time of the polyp in the simulation was fixed at 1 s. Four sets of experiments with different contraction time duty cycles, or expansion/contraction time ratios, were conducted to simulate the expansion and contraction of the polyp subumbrella: 1 s:1 s, 2 s:1 s, 3 s:1 s, and 4 s:1 s, respectively. All sets of simulated movements were implemented separately using the corresponding smoothing functions. As the displacements of the subumbrella of the simulated polyp only occurred in the r-direction, the displacement of the subumbrella of the simulated polyp in the a-direction and φ-direction were prescribed as 0 to avoid unexpected twisting of the mesh and to ensure convergence of the simulations.

In the fluid domain, the *Re* number (approximately 0.96, less than 1) was calculated by using the diameter of the opening of the simulated subumbrella at 0.003 mm within the polyp as the characteristic size. In this case, viscous forces dominated the fluid domain, and the influence of inertial forces was negligible (i.e. the inertial term in the Navier-Stokes equations was equal to zero); thus, the peristaltic flow model was chosen for all simulations. We assumed that the tetraradial pagoda-shaped *Quadrapyrgites* lived in environments with relatively low flow velocities. To better visualise the effect of the dynamics of *Quadrapyrgites* on the surrounding environment, the simulated fluid domain was set as a hydrostatic environment. The top and perimeter of the simulated domain were set as open boundaries to allow the flow of water in and out without restrictions. In addition, the bottom boundary, umbrella surface, and peridermal surface of the polyp in contact with the fluid were set as the no-slip boundaries. The density and dynamic viscosity of water in the simulation were set to 1000 kg/m$^3$ and 0.001 Pa·s, respectively.

The simulations used a unidirectional structure to the fluid coupling method (i.e. velocity transmission to fluid) to couple fluid and solid fields. In addition, the moving mesh feature was enabled, allowing for the subumbrella model inside the periderm to be deformed. Considering the complexity and computational cost of the physical fields in the simulation domain, the governing equations were solved using a time-dependent, non-linear, and fully coupled solver with a relative tolerance of 0.005 and a time step of T=0.01 s for the final output.

To obtain flow velocity data above the peridermal aperture and determine the effective range of the active expansion/contraction of *Quadrapyrgites*, 10 sampling cut points (Cut Point 3D) were set above the peridermal aperture, ranging from z=2.05 mm to z=2.5 mm at the coordinates of the simulated domain, with 0.05 mm between each cut point (*Figure 3—figure supplement 1B*). The results were visualised as 2D cross sections along the centre of the rectangular simulation domain. The simulation files were saved as '.mph' files.

## Mesh sensitivity analysis

For the sensitivity analysis, a mesh model (*Figure 3*) with an expansion/contraction time ratio of 3:1 was used as the test model. Meshes with different levels of refinement were tested, with the number of elements ranging from 75,781 to 1,009,782 (Table supplement 2 in figshare). The flow velocity values obtained from the cut points with different meshes on each timestep were compared, and the average difference of each timestep was calculated. Results were considered mesh independent when the difference ranged from 5% to 10% between those obtained with the current mesh and the next most fine mesh. Finally, the mesh selected for subsequent simulations contained 670,654 elements in total (see Table supplement 3 in figshare for specific mesh parameters). The average error between the results obtained with such meshes and those with a finer mesh was approximately 5.5% (Table supplement 2 in figshare). Four additional simulations for sensitivity analysis of the parameters of boundary layer mesh were also conducted, including the layer number (five and eight layers, respectively) and thickness (controlled by different thickness adjustment factor). Only the thickness of boundary layer mesh can influence the maximum flow velocity of the contraction phase. However, the results of all the four simulations were generally consistent (Table supplement 2 in figshare).

## Acknowledgements

We thank HJ Gong, X Liu, and MR Cheng (State Key Laboratory for Continental Dynamics, Northwest University, Xi'an, China) for their assistance in both field and lab work.

# Additional information

## Funding

| Funder | Grant reference number | Author |
| --- | --- | --- |
| Natural Science Foundation of China | 42372012 | Jian Han |

| Funder | Grant reference number | Author |
|---|---|---|
| Chinese Academy of Sciences | XDB26000000 | Jian Han |
| Ministry of Education of China | D17013 | Jian Han |
| Northwest University | BJ11060 | Jian Han |
| National Key Research and Development Program of China | 2023YFF0803601 | Jian Han |
| Linyi University | Z6122059 | Xing Wang |
| Natural Science Foundation of China | 41720104002 | Jian Han |
| Natural Science Foundation of China | 41911530236 | Jian Han |

The funders had no role in study design, data collection and interpretation, or the decision to submit the work for publication.

## Author contributions

Yiheng Zhang, Software, Visualization, Methodology, Writing – original draft, Writing – review and editing; Xing Wang, Conceptualization, Resources, Formal analysis, Writing – original draft, Writing – review and editing; Jian Han, Conceptualization, Resources, Supervision, Funding acquisition, Validation, Writing – original draft, Writing – review and editing; Juyue Xiao, Writing – original draft, Writing – review and editing; Yuanyuan Yong, Chiyang Yu, Ning Yue, Formal analysis, Writing – original draft, Writing – review and editing; Jie Sun, Resources, Software; Kaiyue He, Wenjing Hao, Formal analysis; Tao Zhang, Conceptualization, Supervision, Validation, Investigation, Writing – original draft, Project administration; Bin Wang, Supervision, Validation, Methodology, Writing – review and editing; Deng Wang, Xiaoguang Yang, Formal analysis, Writing – original draft

## Author ORCIDs

Yiheng Zhang https://orcid.org/0009-0002-2426-0838
Xing Wang https://orcid.org/0000-0002-1777-864X
Jian Han https://orcid.org/0000-0002-2134-4078
Tao Zhang https://orcid.org/0000-0002-5622-0227

Reviewer #1 (Public Review): https://doi.org/10.7554/eLife.90211.4.sa1
Reviewer #2 (Public Review): https://doi.org/10.7554/eLife.90211.4.sa2
Author response https://doi.org/10.7554/eLife.90211.4.sa3

# Additional files

## Supplementary files
MDAR checklist

## Data availability
Supplementary data and simulation files of this work have been deposited in figshare.

The following dataset was generated:

| Author(s) | Year | Dataset title | Dataset URL | Database and Identifier |
|---|---|---|---|---|
| Zhang Y | 2025 | Supplementary files for "Dynamic simulations of feeding and respiration of the early Cambrian periderm-bearing cnidarian polyps" | https://doi.org/10.6084/m9.figshare.23282627 | figshare, 10.6084/m9.figshare.23282627 |

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
