## [Editor Report · eLife assessment]

This **important** study advances our understanding of early Cambrian cnidarian paleoecology and suggests that the reconstructed ancestral feeding and respiration mechanisms predate jet-propelled swimming utilized by modern jellyfish. The work combines **solid** evidence of fluid and structural mechanics modeling, simulating for the first time the feeding and respiratory capacities in a microfossil (Quadrapyrgites), which in turn opens new possibilities using this approach for paleontological research. Assuming that the prior interpretations and assumptions concerning the modeled organism's soft part and skeletal anatomy are correct, the hypotheses that (1) the organism could alternately contract and expand the oral region and (2) such movement increased feeding efficiency seem plausible.

---

## [Referee Report · Reviewer #1 (Public Review)]

Summary:

The authors utilize fluid-structure interaction analyses to simulate fluid flow within and around the Cambrian cnidarian Quadrapyrgites to reconstruct feeding/respiration dynamics. Based on vorticity and velocity flow patterns, the authors suggest that the polyp expansion and contraction ultimately develop vortices around the organism that are like what modern jellyfish employ for movement and feeding. Lastly, the authors suggest that this behavior is likely a prerequisite transitional form to swimming medusae.

Strengths:

While fluid-structure-interaction analyses are common in engineering, physics, and biomedical fields, they are underutilized in the biological and paleobiological sciences. Zhang et al. provide a strong approach to integrating active feeding dynamics into fluid flow simulations of ancient life. Based on their data, it is entirely likely the described vortices would have been produced by benthic cnidarians feeding/respiring under similar mechanisms. However, some of the broader conclusions require additional justification.

Weaknesses:

(1) The claim that olivooid-type feeding was most likely a prerequisite transitional form to jet-propelled swimming needs much more support or needs to be tailored to olivooids. This suggests that such behavior is absent (or must be convergent) before olivooids, which is at odds with the increasing quantities of pelagic life (whose modes of swimming are admittedly unconstrained) documented from Cambrian and Neoproterozoic deposits. Even among just medusozoans, ancestral state reconstruction suggests that they would have been swimming during the Neoproterozoic (Kayal et al., 2018; BMC Evolutionary Biology) with no knowledge of the mechanics due to absent preservation.

(2) While the lack of ambient flow made these simulations computationally easier, these organisms likely did not live in stagnant waters even within the benthic boundary layer. The absence of ambient unidirectional laminar current or oscillating current (such as would be found naturally) biases the results.

(3) There is no explanation for how this work could be a breakthrough in simulation gregarious feeding as is stated in the manuscript.

Despite these weaknesses the authors dynamic fluid simulations convincingly reconstruct the feeding/respiration dynamics of the Cambrian Quadrapyrgites, though the large claims of transitionary stages for this behavior are not adequately justified. Regardless, the approach the authors use will be informative for future studies attempting to simulate similar feeding and respiration dynamics.

---

## [Referee Report · Reviewer #2 (Public Review)]

Summary:

The authors seek to elucidate the early evolution of cnidarians through computer modeling of fluid flow in the oral region of very small, putative medusozoan polyps. They propose that the evolutionary advent of the free-swimming medusoid life stage was preceded by a sessile benthic life stage equipped with circular muscles that originally functioned to facilitate feeding and that later became co-opted for locomotion through jet propulsion.

Strengths:

Assumptions of the modeling exercise laid out clearly; interpretations of the results of the model runs in terms of functional morphology plausible. An intriguing investigation that should stimulate further discussion and research.

Weaknesses:

Speculation on the origin of the medusoid life stage in cnidarians heavily dependent on prior assumptions concerning the soft part anatomy and material properties of the skeleton of the modeled fossil organism that may be open to alternative interpretations. Logically, of course, the hypothesis that cnidarian medusae originated from benthic polyps must be evaluated along with the alternative hypotheses that the medusa came first and that the ancestral cnidarian exhibited both life stages.

---

## [Author Response]

The following is the authors’ response to the previous reviews.

**Reviewer #1 (Public Review):**
Original comment: There is no explanation for how this work could be a breakthrough in simulation gregarious feeding as is stated in the manuscript.Reviewer response: I think I understand where the authors are trying to take this next step. If the authors were to follow up on this study with the proposed implementation of inhalant/exhalent velocities profiles (or more preferably velocity/pressure fields), then that study would be a breakthrough in simulating such gregarious feeding. Based on what has been done within the present study, I think the term "breakthrough" is instead overly emphatic. An additional note on this. The authors are correct that incorporating additional models could be used to simulation a population (as has been successfully done for several Ediacaran taxa despite computational limitations), but it's not the only way. The authors 1 might explore using periodic boundary conditions on the external faces of the flow domain. This could require only a single Olivooid model to assess gregarious impacts - see the abundant literature of modeling flow through solar array fields.

We appreciate the reviewer 1 for the suggestion. Modeling gregarious feeding via periodic boundary conditions is surely a practical way with limited computational resources. Modeling flow through solar array fields can also be an inspiring case. However, to realism the simulation of gregarious feeding behavior on an uneven seabed and with irregular organism spatial distribution, just using periodic boundary conditions may not be sufficient (see Author response image 1 for a simple example). We will go on exploring the way of realizing the simulations of large-scale gregarious feeding.

**Author response image 1. sa3fig1:** An example of modeling gregarious feeding behavior on an uneven seabed.

Original comment: The claim that olivooid-type feeding was most likely a prerequisite transitional form to jet-propelled swimming needs much more support or needs to be tailored to olivooids. This suggests that such behavior is absent (or must be convergent) before olivooids, which is at odds with the increasing quantities of pelagic life (whose modes of swimming are admittedly unconstrained) documented from Cambrian and Neoproterozoic deposits. Even among just medusozoans, ancestral 1 state reconstruction suggests that they would have been swimming during the Neoproterozoic (Kayal et al., 2018; BMC Evolutionary Biology) with no knowledge of the mechanics due to absent preservation. Author response: Thanks for your suggestions. Yes, we agree with you that the ancestral swimming medusae may appear before the early Cambrian, even at the Neoproterozoic deposits. However, discussions on the affinities of Ediacaran cnidarians are severely limited because of the lack of information concerning their soft anatomy. So, it is hard to detect the mechanics due to absent preservation. Olivooids found from the basal Cambrian Kuanchuanpu Formation can be reasonably considered as cnidarians based on their radial symmetry, external features, and especially the internal anatomies (Bengtson and Yue 1997; Dong et al. 2013; 2016; Han et al. 2013; 2016; Liu et al. 2014; Wang et al. 2017; 2020; 2022). The valid simulation experiment here was based on the soft tissue preserved in olivooids.Reviewer response: This response does not sufficiently address my earlier comment. While the authors are correct that individual Ediacaran affinities are an area of active research and that Olivooids can reasonably be considered cnidarians, this doesn't address the actual critique in my comment. Most (not all) Ediacaran soft-bodied fossils are considered to have been benthic, but pelagic cnidarian life is widely acknowledged to at least be present during later White Sea and Nama assemblages (and earlier depending on molecular clock interpretations). The authors have certainly provided support for the mechanics of this type of feeding being co-opted for eventual jet propulsion swimming in Olivooids. They have not provided sufficient justifications within the manuscript for this to be broadened beyond this group.

Thanks for your sincere commentary. We of course agree with the possibility of the emergence of swimming cnidarians before the lowermost Cambrian Fortunian Stage. See lines 16-129: “Ediacaran fossil assemblages with complex ecosystems consist of exceptionally preserved soft-bodied eukaryotes of enigmatic morphology, which their affinities are mostly unresolved (Tarhan et al., 2018, Integrative and Comparative Biology, 58 (4), 688–702; Evans et al., 2022, PNAS, 11(46), e220747511).” Undoubtedly Olivooids belong to cnidarians charactered by their external and internal biological structures. Limited by the fossil records, we could only speculate on the transition from the benthic to the swimming of ancestral cnidarians via the valid fossil preservation, e.g. olivooids. The transition may require processes such as increasing body size, thickening the mesoglea, and degenerating the periderm, etc. And these processes may also evolve independently or comprehensively. Moreover, the ecological behaviors of the ancestral cnidarians may evolve independently at different stages from Ediacaran to Cambrian. We therefore could not provide more sufficient justifications beyond olivooids.

Original comment: L446: two layers of hexahedral elements is a very low number for meshing boundary layer flowReviewer response: As the authors point out in the main text, these organisms are small (millimeters in scale) and certainly lived within the boundary layer range of the ocean. While the boundary layer is not the main point, it still needs to be accurately resolved as it should certainly affect the flow further towards the far field at this scale. I'm not suggesting the authors need to perfectly resolve the boundary layer or focus on using turbulence models more tailored to boundary layer flows (such as k-w), but the flow field still needs sufficient realism for a boundary bounded flow. The authors really should consider quantitatively assessing the number of hexahedral elements within their mesh refinement study.

To address this concern, we run another four simulations based on mesh4 within our mesh refinement study to assess the number of hexahedral elements (five layers and eight layers of hexahedral elements with different thickness of boundary layer mesh (controlled by thickness adjustment factor), respectively). the results had been supplemented to Table supplement 2. As shown in the results, the number of layers of hexahedral elements seems does not significant influence the result, but the thickness of boundary layer mesh can influence the maximum flow velocity of the contraction phase. However, the results of all the simulations were generally consistent, as shown in Author response image 2. The description of the results above were added to section “Mesh sensitivity analysis”.

**Author response image 2. sa3fig2:** Results of mesh refinement study of different boundary layer mesh parameters.